# Differentiated Local Differential Privacy Bloom Filter for Membership Queries

## Abstract

We study the problem of privacy-preserving membership queries over large-scale datasets, where Bloom filters are widely adopted for their efficiency and scalability. Existing approaches, however, employ a fixed number of hash functions for all elements, overlooking their varying importance or frequency. This uniform treatment induces a suboptimal balance between privacy and utility: frequent or critical elements demand more accurate encoding and finely tuned protection, while less significant elements can tolerate higher uncertainty without severely affecting overall performance.

Our main technical result is a Differentiated Local Differential Privacy Bloom Filter for membership queries (DLDP-BF), which dynamically allocates hash functions and privacy budgets according to element importance, significantly improving query accuracy for data. Moreover, we design a novel LDP budget allocation algorithm that adaptively adjusts noise intensity in proportion to element importance. We further establish a mathematical model linking importance and privacy budget allocation, and provide a theoretical analysis proving that our method ensures strict local differential privacy guarantees while balancing data utility. Our experiments demonstrate the favourable performance of our approach in real-world settings, and highlight the data utility benefits of the privacy-preserving membership queries problem.

## 1 Introduction

Membership query is a fundamental operation that checks whether a given element exists in a specific dataset Angluin & Kharitonov (1991); Tarkoma et al. (2011); Diakonikolas et al. (2024), and is widely used in tasks such as blacklist detection, genomic analysis, and contact discovery Fan et al. (2024); Engels et al. (2021); Filić et al. (2024). To efficiently support membership queries in large-scale systems, the Bloom Filter (BF) has emerged as a classical and widely adopted solution Bloom (1970). Due to its simplicity and efficiency, the Bloom Filter has become a fundamental and indispensable tool for membership query tasks. It is extensively applied across various domains, including databases, network security, and distributed systems Liu et al. (2020); Geravand & Ahmadi (2013); Luo et al. (2018).

However, membership queries based on Bloom filters often face serious security and privacy risks. For example, in contact discovery services, users are often required to upload their entire address book to query which contacts are registered on a platform. This approach exposes all of a user's social relationships to the platform. Moreover, it enables the inference of user behavior patterns and social network structures through data mining, posing a serious threat to personal privacy Demmler et al. (2018); Yu et al. (2024). If effective protection mechanisms are lacking, attackers could analyze query content or response results to infer whether a dataset contains specific sensitive elements, or even infer portions of the original data, leading to privacy leaks Hu et al. (2023). Therefore, when providing such query services, data owners must implement practical privacy protection measures to minimize the risk of sensitive information leakage Tang et al. (2016); Guan et al. (2024).

To address this challenge, local differential privacy (LDP) Kasiviswanathan et al. (2011) provides a powerful paradigm for protecting sensitive information in membership queries by applying random perturbations on the user side. The LDP Bloom filter for membership query is illustrated in Fig. 1. Despite some research progress, existing LDP Bloom filter member query methods typically

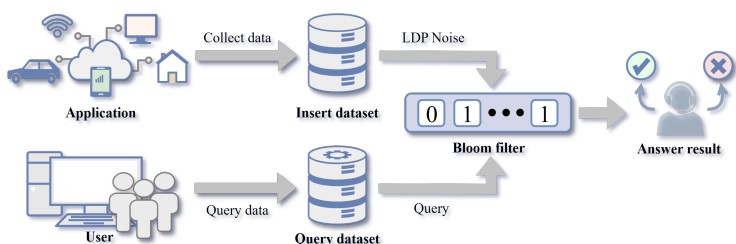

Figure 1: The LDP Bloom filter for membership query.

employ a uniform parameter configuration strategy, assigning the same number of hash functions and a uniform privacy budget to all elements, which may lead to a decrease in accuracy in practical applications. Specifically, existing methods face the following two key technical challenges:

(1) Non-Differentiated Hash Function Assignment. Existing methods for differentially private membership queries based on Bloom filters assign the same number of hash functions to all data items, without considering the varying importance of data. However, in real-world scenarios, data distribution often exhibits non-uniformity. Because the number of hash functions directly affects the false positive rate and query efficiency of Bloom filters, a uniform hash function allocation strategy struggles to balance the varying importance of data items and may result in wasted resources on less important data.

(2) Non-Differentiated Privacy budget allocation. Existing methods often allocate the same privacy budget to all data items, regardless of their varying importance. This uniform allocation neglects the fact that both the privacy budget and Bloom filter parameters jointly determine the level of perturbation and query accuracy. As a result, less important data may be overprotected while critical data may receive insufficient protection, leading to degraded overall data utility and suboptimal system performance.

**Our Contribution.** We propose a Differentiated Local Differential Privacy Bloom Filter for Membership Queries (DLDP-BF) to achieve the privacy-preserving membership query problem for elements with different element importance. Summary of our contributions are:

• We propose a DLDP-BF that dynamically adjusts both hash function assignment and privacy budget based on each element's importance. Important elements—those with higher query frequency or membership probability—are assigned more hash functions and larger privacy budgets to reduce false positives and enhance utility, while less important elements receive fewer resources to reduce overhead. To our knowledge, the proposed Personalized Budget Allocation under Local Differential Privacy (PBA) is the first local differential privacy budget allocation method that jointly considers membership probability and query frequency.

• The proposed method achieves a time complexity of $O(m \ln 2 \cdot (\mathcal{T}_H + 1))$, ensuring high computational efficiency and scalability for large-scale datasets. A theoretical privacy analysis demonstrates that the method satisfies local differential privacy, providing strong guarantees against inference attacks. The utility analysis further shows that adaptive allocation of computational and privacy resources preserves high query accuracy while maintaining strong privacy protection, thus achieving an effective balance between usability and security in membership query scenarios.

• Extensive experiments conducted on multiple diverse datasets demonstrate the effectiveness of the proposed algorithm. The results show that the method not only provides strong privacy protection but also significantly improves data utility and query accuracy in practical membership query scenarios. The method achieves varying degrees of improvement under different privacy budgets $\epsilon$ and bit array sizes $m$. Specifically, it attains an average reduction in root mean square error (RMSE) of 37.1% and an average accuracy improvement of 9.05%, highlighting its ability to effectively balance privacy preservation and data utility.

## 2 RELATED WORK

Recent research has begun exploring how to combine Bloom filters with Local Differential Privacy (LDP) to achieve privacy-preserving membership queries. The general process involves first encoding elements into a Bloom filter, then perturbing the bit array using a local or centralized differential privacy mechanism, and finally performing membership queries on the resulting noisy structure. However, research on Bloom filter membership query methods for LDP remains limited. RAPPOR (Randomized Aggregatable Privacy-Preserving Ordinal Response) Erlingsson et al. (2014) applies permanent and immediate random responses to the client-encoded Bloom filter bits, achieving strong local privacy. Ke et al. Ke et al. (2025) proposed DPBloomFilter, which directly injects LDP-compliant noise into the Bloom filter bit array. This method improves query accuracy while maintaining strict local differential privacy through global parameter tuning and noise injection.

Another important research direction focuses on adaptive privacy budget allocation to improve utility in LDP mechanisms. Jia and Gong Jia & Gong (2019) proposed a distribution-aware method for frequency estimation and heavy hitter identification, leveraging prior knowledge of the underlying frequency distribution to guide perturbation. Shen et al. Shen et al. (2021) proposed PLDP for multidimensional data, allowing each user or dimension to have a personalized privacy budget. Data are perturbed according to the individual budget before aggregation, improving accuracy compared with uniform LDP mechanisms. Wei et al. Wei et al. (2024) proposed the Advanced Adaptive Additive (AAA) mechanism for mean estimation under LDP. It first samples a small subset of users to construct a noisy data descriptor, and then perturbs the remaining data in a distribution-aware manner, optimizing the perturbation with respect to task-specific utility.

Existing research demonstrates that distribution-aware design and adaptive budget allocation can improve the utility of LDP mechanisms. Inspired by these studies, this work integrates differentiated hash function allocation with personalized privacy budgets into a Bloom filter structure. By leveraging prior distribution information, the proposed method optimizes the accuracy of membership queries while satisfying strict LDP constraints.

## 3 PRELIMINARIES

### 3.1 BLOOM FILTER

A Bloom filter Bloom (1970) is a space-efficient probabilistic data structure used to test whether an element is a member of the set. Bloom filter-based encoding is an efficient and widely used technique, originally for strings and categorical data Vatsalan & Christen (2014); Schnell et al. (2009); Hancock & Khoshgoftaar (2020), and recently extended to numerical data Vatsalan & Christen (2016). Its formal definition is as follows.

**Definition 1** (Bloom Filter Bloom (1970)). *A Bloom filter is used to represent a set $S = \{x_1, x_2, \ldots, x_n\}$ of $|S|$ elements from a universe $U$. A Bloom filter consists of a binary array $g \in \{0,1\}^m$ of $m$ bits, which are initially all set to 0, and uses $l$ independent random hash functions $h_1, \ldots, h_l$ with range $\{0, \ldots, m-1\}$. These hash functions map each element in the universe to a random number uniform over the range $\{0, \ldots, m-1\}$ for mathematical convenience. The computation time per execution for all hash functions is $\mathcal{T}_H$.*

### 3.2 LOCAL DIFFERENTIAL PRIVACY

Local differential privacy (LDP) Kasiviswanathan et al. (2011) is a privacy model that ensures each individual's data is randomized locally on the user device, such that even if an adversary accesses the raw data, they cannot reliably infer additional personal information Evfimievski et al. (2003); Lyu (2022). Unlike traditional centralized differential privacy, LDP protects inputs before they leave the device. In recent years, LDP has become a widely adopted standard, with implementations by major technology companies such as Google and Apple Erlingsson et al. (2014); Kim et al. (2018); Truex et al. (2020).

**Definition 2** ($\epsilon$-Local Differential Privacy Kasiviswanathan et al. (2011)). *A randomized algorithm $\mathcal{M}$ satisfies $\epsilon$-local differential privacy (LDP) if for any two inputs $v_1$ and $v_2$, and any possible output $y$, the following holds: $\Pr(\mathcal{M}(v_1) = y) \leq e^{\epsilon} \cdot \Pr(\mathcal{M}(v_2) = y)$, where $\epsilon$ is the privacy budget, controlling the level of privacy.*

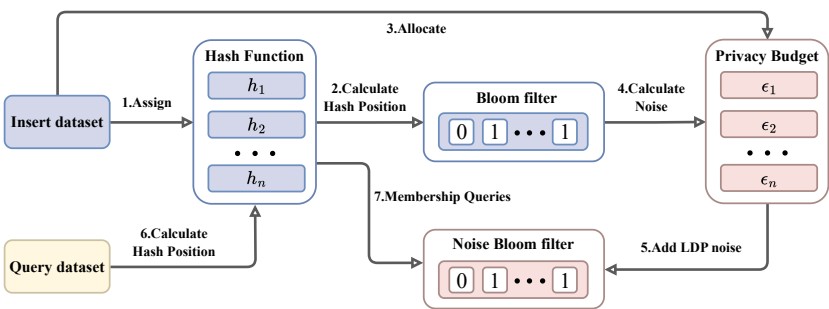

Figure 2: The overview of the DLDP-BF Workflow.

### 3.3 MEMBERSHIP LIKELIHOOD AND QUERY FREQUENCY

**Definition 3** (Membership Likelihood Bruck et al. (2006); Zhong et al. (2008); Fan et al. (2022)). *The membership likelihood of an element $i \in U$ is defined as the probability that $i$ belongs to the set $S$, providing a probabilistic view of set membership. Each element $i$ is associated with an indicator random variable $X_i$, defined as $X_i = 1$ if $i \in S$ and $X_i = 0$ if $i \notin S$. The probability that $i$ belongs to $S$ is denoted by $x_i$, i.e., $P(X_i = 1) = x_i$, where $x_i \in (0, 1)$.*

**Definition 4** (Query Frequency Bruck et al. (2006); Zhong et al. (2008); Fan et al. (2022)). *The query frequency is defined as the probability that an element $i \in U$ is queried by the client during a single query request, reflecting the likelihood of access to a specific element. The specific definition is as follows. Each element $i$ is associated with an indicator random variable $Y_i$, defined as $Y_i = 1$ if $i$ is queried, and $Y_i = 0$ otherwise. The probability that element $i$ is queried is denoted by $y_i$, i.e., $P(Y_i = 1) = y_i$, where $y_i \in (0, 1)$.*

Note that the parameters $X_i$ and $Y_i$, which are used to guide personalized perturbation in privacy-preserving mechanisms, can be safely estimated from historical or aggregated statistics. This approach ensures that the computation of these parameters does not directly expose any individual's sensitive data while providing sufficient information for algorithmic optimization Bruck et al. (2006); Zhong et al. (2008); Fan et al. (2022).

## 4 METHODOLOGY

In this section, we first provide an overview of the workflow in Section 4.1. We then describe the implementation details of the Differentiated Hash Function Assignment (DHFA) algorithm in Section 4.2, followed by the Privacy Budget Allocation for Local Differential Privacy (PBA) algorithm in Section 4.3.

### 4.1 OVERVIEW

In this section, we provide an overview of DLDP-BF, a Differentiated Local Differential Privacy Bloom Filter for Membership Queries, designed to offer differential privacy protection for elements with different importance data.

To clearly define the privacy protections in our work, we employ a standard local threat model Kasiviswanathan et al. (2011); Dwork et al. (2006); Erlingsson et al. (2014): an untrusted but "honest but curious" server that can only observe perturbed member query results and cannot access the raw data. Our DLDP-BF design ensures that even if the server possesses auxiliary information, but not its internal parameters, it cannot confidently determine whether any particular element belongs to the target dataset.

The overview of the DLDP-BF workflow is illustrated in Fig. 2. The core idea behind DLDP-BF is its adaptive approach to both hash function selection and privacy budget allocation. Firstly, the number of hash functions is dynamically adjusted based on the importance of each data element, improving query accuracy. Secondly, DLDP-BF allocates privacy budgets precisely according to

the membership probability and query frequency of elements, ensuring privacy is preserved while minimizing noise. This design balances privacy protection and query efficiency, minimizing the impact on data utility while adhering to local differential privacy requirements. The detailed process and implementation steps of DLDP-BF are outlined in Algorithm 1.

---

**Algorithm 1:** Differentiated Local Differential Privacy Bloom Filter for Membership Queries

---

**Input:** $m$: m-bit array initialized to all zeros; $n$: expected number of elements in $S \subseteq U$; $\mathcal{F}_i$: normalized query frequency of $i \in U$; $\mathcal{L}_i$: membership likelihood of $i \in U$; $\{H_1, \ldots, H_l\}$: independent hash functions; $Q$: query sequence $Q = \{q_1, q_2, \ldots\}, q_j \in U$; $\epsilon$: privacy budget.

**Output:** Membership result set $R = \{r_1, r_2, \ldots, r_{|Q|}\}$, where $r_i \in \{0, 1\}$.

1 Initialize BF $\leftarrow [0]^m$ ;                                    // Bloom Filter array
2 Initialize $R \leftarrow \emptyset$ ;                                    // Store query results
3 **Inserting Phase:**
4 **for** *each element $x \in S$* **do**
5     BF $\leftarrow$ DHFA($x$) ;   // Assign hash functions and set corresponding Bloom Filter bits for element $x$ (Alg 2)
6     $\tilde{\text{BF}} \leftarrow$ PBA($x$, BF) ;       // Perturb Bloom Filter bits for $x$ using personalized LDP budget (Alg 3)
7 **end**
8 **Querying Phase:**
9 **for** *each query $q_i \in Q$* **do**
10     $h_i^* \leftarrow$ DHFA($q_i$) ;       // Use pre-computed hash function $h_i^*$ for $q_i$
11     $r_i \leftarrow 1$ ;                   // Initialize membership result for $q_i$ to 1
12     **for** $j \leftarrow 0$ **to** $h_i^* - 1$ **do**
13        **if** $\tilde{BF}[H_j(q_i)] == 0$ **then**
14           $r_i \leftarrow 0$
15           break ; // If any corresponding bit is 0, element is not in the set; otherwise remains 1
16        **end**
17     **end**
     // Store the membership result
18     $R$.append($r_i$)
19 **end**
20 **return** Membership result set $R$

---

## 4.2 DIFFERENTIATED HASH FUNCTION ASSIGNMENT ALGORITHM (DHFA)

---

**Algorithm 2:** Differentiated hash function assignment algorithm (DHFA)

---

**Input:** $m$: m-bit array initialized to all zeros; $n$: expected number of elements in set $S \subseteq U$; $\mathcal{F}_i$: normalized query frequency of element $i \in U$; $\mathcal{L}_i$: membership likelihood of element $i \in U$; $H$: hash function family $\{H_0, H_1, ..., H_l\}$.

**Output:** Bloom Filter BF.

1 Initialize BF $\leftarrow [0]^m$ ;                                    // Bloom Filter bit array
2 **for** *each element $x \in S$* **do**
3     $h_i^* \leftarrow \frac{m}{n} \ln 2 + \log_2\left(\frac{\mathcal{F}_i}{\mathcal{L}_i}\right) - \sum_{j \in U} \frac{\mathcal{L}_j}{n} \log_2\left(\frac{\mathcal{F}_j}{\mathcal{L}_j}\right)$ ;   // Compute optimal hash count
4     **for** $j \leftarrow 0$ **to** $h_i^* - 1$ **do**
5        BF$[H_j(x)] \leftarrow 1$ ;                   // Flip bit via hash function $H_j$
6     **end**
7 **end**
8 **return** Bloom Filter BF

---

The classical LDP Bloom filter for membership query, despite its widespread adoption, is inherently limited by its static parameter configuration. It presumes a uniform distribution of both query frequencies and membership probabilities across elements—an assumption that often fails in practical applications. Consequently, its performance can degrade significantly in the presence of different important elements. To address the limitation, we propose a Differentiated Hash Function Assignment algorithm (DHFA), designed to dynamically adjust the allocation of hash functions based on prior knowledge of query frequency and membership probability distributions. The detailed process and implementation steps of PBA are shown in Algorithm 2.

In many real-world applications, query frequencies and membership probabilities can be inferred through statistical profiling. DHFA leverages this information to prioritize high-impact elements—commonly referred to as *hot items*—by assigning them proportionally greater resources.

To formalize this algorithm, we derive an optimal parameter configuration for the Bloom Filter using a probabilistic modeling approach. Specifically, we integrate the distributions of query frequency and membership likelihood into a unified framework that assigns different numbers of hash functions to different elements. When the query frequency distribution is uniform, the derived configuration simplifies to that of the standard Bloom filter, highlighting the classical design as a special case within this broader formulation.

Let the expected size of the set be $n$, and assume a Bloom filter with an $m$-bit array. The total number of hash functions is given by $K^* = m \ln 2$. For each element, the optimal number of hash functions is determined by:

$$h_i^* = \frac{m}{n} \ln 2 + \log_2 \left( \frac{\mathcal{F}_i}{\mathcal{L}_i} \right) - \sum_{j \in U} \frac{\mathcal{L}_j}{n} \log_2 \left( \frac{\mathcal{F}_j}{\mathcal{L}_j} \right). \tag{1}$$

In summary, the proposed Differentiated Hash Function Assignment algorithm (DHFA) generalizes the classical LDP Bloom filter membership query method to effectively accommodate data items with varying degrees of importance. By incorporating both query frequency and membership probability into its parameterization, our algorithm achieves improved efficiency, reduced false positive rates, and enhanced applicability in real-world systems.

### 4.3 PERSONALIZED BUDGET ALLOCATION UNDER LOCAL DIFFERENTIAL PRIVACY (PBA)

---

**Algorithm 3:** Personalized Budget Allocation under Local Differential Privacy (PBA)

**Input:** $S$: subset $S \subseteq U$ sampled from distributions $X_i, i \in U$; $\mathcal{F}_i$: normalized query frequency of element $i \in U$; $\mathcal{L}_i$: membership likelihood of element $i \in U$; $h_i^*$: pre-computed optimal hash count for the $i$-th element $x_i$; $\epsilon$: privacy budget.

**Output:** Noise Bloom Filter $\tilde{\text{BF}}$.

1 **for** *each element* $x \in S$ **do**
2    $\epsilon_i^* \leftarrow \epsilon + \log_2 \left( \frac{\mathcal{F}_i}{\mathcal{L}_i} \right) - \sum_{j \in U} \frac{\mathcal{L}_j}{n} \log_2 \left( \frac{\mathcal{F}_j}{\mathcal{L}_j} \right)$ ;      // Compute personalized privacy budget
3    $b \leftarrow \frac{\epsilon_i^*}{\epsilon_i^* + 1}$ ;          // Bernoulli flip probability
4    **for** $j \leftarrow 0$ **to** $h_i^* - 1$ **do**
5      $r \leftarrow \text{U}([0, 1])$ ;          // Generate random flip decision
6      **if** $r > b$ **then**
7        $\text{BF}[H_j(i)] \leftarrow 1 - \text{BF}[H_j(i)]$ ;      // Bit flipping for LDP
8      **end**
9    **end**
10 **end**
11 **return** Noise Bloom Filter $\tilde{\text{BF}}$

---

In order to enhance the accuracy of privacy-preserving membership queries, we introduce the Personalized Budget Allocation under Local Differential Privacy (PBA). The detailed process and implementation steps of PBA are shown in Algorithm 3. This algorithm dynamically adjusts the pri-

vacy budget allocation based on two key factors: the query frequency of each element and its probability of membership in the set. By incorporating these characteristics, the algorithm provides a personalized privacy protection strategy that tailors the privacy budget to the significance of each element.

In traditional methods, privacy budgets are typically fixed or statically allocated across all elements, regardless of their usage patterns or importance. This fixed approach can lead to inefficiencies, especially in scenarios where certain elements are queried more frequently or are of greater importance. By contrast, the PBA algorithm adapts to these variations by allocating more privacy budget to high-frequency queried elements, thereby offering more precise and effective privacy protection.

The dynamic allocation of the privacy budget ensures that elements with higher query frequencies receive more robust privacy guarantees, which in turn helps reduce the false positive rate of membership queries. By optimizing element-specific parameters, this mechanism improves query accuracy while providing stronger privacy guarantees. Through this adaptive strategy, the algorithm strikes a better balance between privacy protection and data usability, ensuring that the privacy protection mechanism does not excessively degrade the quality of the results.

As illustrated in Equation 2, the Privacy Budget Allocation Algorithm (PBA) computes the optimal privacy budget allocation for each element based on its query frequency and membership probability. By incorporating the importance of each element, the algorithm adjusts the privacy budget allocation in a way that ensures fairness and accuracy for all elements while adhering to the principles of local differential privacy. This dynamic adjustment allows for a more efficient use of the privacy budget, ensuring that each element receives the most appropriate level of privacy protection.

$$\epsilon_i^* = \epsilon + \log_2\left(\frac{\mathcal{F}_i}{\mathcal{L}_i}\right) - \sum_{j \in U} \frac{\mathcal{L}_j}{n} \log_2\left(\frac{\mathcal{F}_j}{\mathcal{L}_j}\right) \tag{2}$$

In this Equation 2, $\epsilon_i^*$ represents the optimal privacy budget for the $i$-th element, where $\epsilon$ is the overall privacy budget, $\mathcal{F}_i$ is the query frequency of the $i$-th element, and $\mathcal{L}_j$ denotes the probability of the $j$-th element being a member of the set. The summation term accounts for the distribution of membership probabilities across all elements. This approach ensures that the privacy budget is allocated in a way that is both fair and efficient, offering a refined balance between protecting sensitive information and maintaining high-quality query results.

Through the PBA algorithm, we aim to significantly improve the effectiveness of privacy-preserving systems, especially in contexts where data access patterns are uneven and certain elements are queried more frequently. This approach enables more granular control over privacy budget allocation, facilitating the development of more efficient and scalable privacy-preserving systems.

## 5 THEORETICAL ANALYSIS

### 5.1 RUNTIME ANALYSIS

**Theorem 1** (The Running Time of DLDP-BF). *Let $\mathcal{T}_H$ denote the time complexity of the hash function operations in our algorithm, and $m$ denote the size of the Bloom Filter bit array, The overall running time of the DLDP-BF is $O\big(m \ln 2 \cdot (\mathcal{T}_H + 1)\big)$.*

*Proof.* We now analyze the overall running time of the algorithm. Considering both the insertion cost and the query cost together, we obtain that the running time of our entire algorithm is $O(m \ln 2 \cdot (\mathcal{T}_H + 1))$. The complete proof of Theorem 1 is provided in Appendix A.1. □

### 5.2 SPACE COMPLEXITY ANALYSIS

**Theorem 2** (Space Complexity of DLDP-BF). *Let $n$ denote the total number of elements and $m$ denote the size of the Bloom Filter bit array. The overall space required by DLDP-BF is $O(m)$.*

*Proof.* The proof follows by showing that the Bloom filter bit array dominates all other components, since the expected total number of hash evaluations equals $m \ln 2$ and the storage for $\mathcal{L}_i$ and $\mathcal{F}_i$ is negligible compared to the $m$-bit array, thus yielding an overall space complexity of $O(m)$. The complete proof of Theorem 2 is provided in Appendix A.2. The complete proof of Theorem 2 is provided in Appendix A.2. $\qquad\square$

### 5.3 Privacy analysis

**Theorem 3** (Local Differential Privacy of Permanent Randomized Response). *The Personalized Budget Allocation under Local Differential Privacy (PBA) satisfies $\epsilon_\infty$-Local differential privacy, where $\epsilon_\infty = h_1^*\epsilon_1 + h_2^*\epsilon_2$, and $h_1^*, h_2^*$ are the number of hash functions used for any adjacent values $v_1$ and $v_2$, respectively, while $\epsilon_1$ and $\epsilon_2$ are their corresponding privacy budgets.*

*Proof.* Consider any pair of adjacent values $v_1, v_2$, with their corresponding Bloom filter positions denoted as $S_1$ and $S_2$. The two sets differ in at most $h_1^* + h_2^*$ bits. For each differing bit, after applying the permanent randomized response, the conditional probability ratio of being 1 is bounded by $e^{\epsilon_1}$ or $e^{\epsilon_2}$ (depending on whether it is mapped by $v_1$ or $v_2$). Multiplying these ratios over all differing bits yields an overall bound of $e^{h_1^*\epsilon_1 + h_2^*\epsilon_2}$ for the perturbed vector $B'$. Hence, the mechanism satisfies $\epsilon_\infty$-local differential privacy with $\epsilon_\infty = h_1^*\epsilon_1 + h_2^*\epsilon_2$. In the special case where $h_1^* = h_2^* = h$ and $\epsilon_1 = \epsilon_2 = \epsilon$, this reduces to the classical uniform noise setting $\epsilon_\infty = 2h\epsilon$. Detailed derivations are provided in Appendix A.3. $\qquad\square$

### 5.4 Utility analysis

**Theorem 4** (Accuracy for Query). *The probability that $\tilde{y}$ equals $y$ satisfies:*

$$\Pr[\tilde{y} = y] \geq \alpha(1 - t - t^{h_i^*}) \left(1 - e^{-2\sum_{i=1}^n h_i^*/m}\right)^{h_i^*} + \alpha t.$$

*Proof.* We first derive the lower bound of the query accuracy for DHFA by analyzing the probability of a bit remaining unset, showing that its false positive rate is bounded by an exponential function, thereby providing an overall accuracy guarantee. Subsequently, after introducing the randomized response mechanism, we analyze the probability that the perturbed query result remains consistent with the original output, and incorporate the prior probability $\alpha$ and the privacy budget factor $t = \frac{e^\epsilon}{e^\epsilon+1}$ to obtain the lower bound of the query accuracy for DLDP-BF. Detailed derivations and results are provided in Appendix A.4. $\qquad\square$

## 6 Experimental Evaluation

### 6.1 Experimental setup

We follow the prior work Erlingsson et al. (2014); Ke et al. (2025) and use the following setup.

**Datasets.** We conduct our experiments on five real-world datasets for the differential privacy membership query problem: the Yellow Taxi Trip dataset, the UK Car Accident dataset, and the Obesity Level Prediction dataset. Further details about these datasets are provided in Appendix B.1.

**Comparison.** For comparison, we evaluate four methods: Non-Privacy Bloom (1970) as a baseline without privacy protection, RAPPOR Erlingsson et al. (2014), which applies randomized-response-based local differential privacy, DPBloomFilter Ke et al. (2025) that injects noise based on correlated sensitivity, and our proposed DLDP-BF, which adaptively allocates hash functions and privacy budgets to achieve high utility under local differential privacy guarantees. Further details about these methods are provided in Appendix B.2.

**Metrics.** In our experimental evaluation, we adopt two primary performance metrics to assess the utility of the differentially private mechanisms: Root Mean Square Error (RMSE) and Accuracy. Further details about these performance metrics are provided in Appendix B.3.

**Environment.** All implementations are developed in Python 3.11 and executed on a workstation running Windows 11, with an AMD Ryzen 7 7735H CPU (3.2 GHz) and 16 GB RAM. Each experiment is repeated 100 times to mitigate randomness, and we report the average metrics to ensure robustness of the results.

## 6.2 PERFORMANCE EVALUATION

### 6.2.1 THE DATA USABILITY OF DIFFERENTIAL PRIVACY BUDGET $\epsilon$

We evaluate the effect of varying privacy budget values ($\epsilon$) on RMSE and accuracy, as shown in Fig. 3 and Fig.4. The results indicate that DLDP-BF consistently outperforms RAPPOR and DP-BloomFilter, providing lower RMSE and higher accuracy across all settings. Specifically, compared to the state-of-the-art differential privacy methods, DLDP-BF achieves a **49.0% reduction in RMSE** and a **12.3% improvement in accuracy**, while approaching the performance of the *Non_Privacy* baseline. These results highlight the strong privacy–utility trade-off enabled by DLDP-BF, with detailed experimental analyses provided in Appendix C.1.

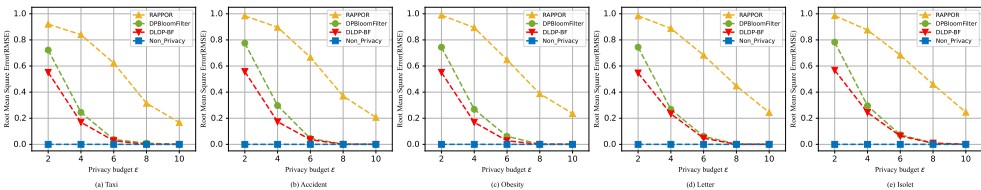

Figure 3: The RMSE of differential privacy budget $\epsilon$

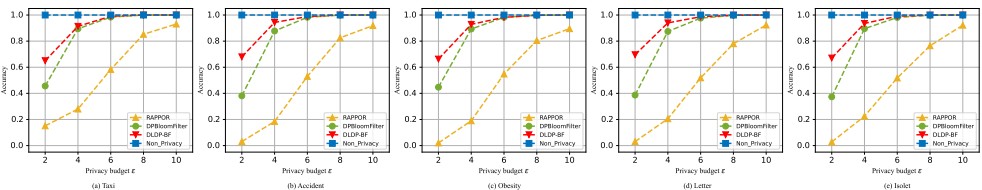

Figure 4: The Accuracy of differential privacy budget $\epsilon$

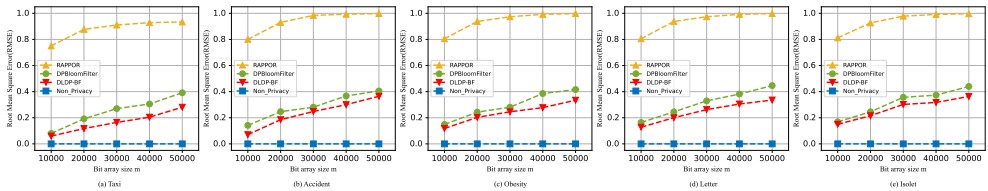

Figure 5: The RMSE of differential Bit array size $m$

### 6.2.2 THE DATA USABILITY OF DIFFERENTIAL BIT ARRAY SIZE $m$

We evaluate the effect of varying bit array sizes ($m$) on RMSE and accuracy, as shown in Fig. 5 and Fig. 6. The results indicate that DLDP-BF consistently outperforms RAPPOR and DPBloomFilter, providing lower RMSE and higher accuracy across all settings. Specifically, compared to the state-of-the-art differential privacy methods, DLDP-BF achieves a **25.2% reduction in RMSE** and a **5.8% improvement in accuracy**, while remaining much closer to the *Non_Privacy* baseline. These results highlight the strong privacy–utility trade-off enabled by DLDP-BF, with detailed experimental analyses provided in Appendix C.2.

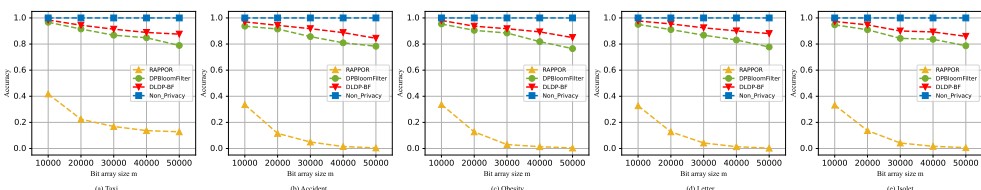

Figure 6: The Accuracy of differential Bit array size $m$

## 7 CONCLUSION

We propose the Differentiated Local Differential Privacy Bloom Filter for Membership Queries(DLDP-BF), which overcomes the limitations of traditional Differential Privacy Bloom filter methods that apply the same fixed hash functions to all elements regardless of importance, resulting in a suboptimal privacy–utility trade-off. We design a differentiated hash function assignment algorithm that deals with different important elements to assign a hash function. Moreover, we propose a privacy budget allocation algorithm that enables differentiated budget distribution, injecting noise of varying magnitudes across different data. Experiments on real datasets show that our method achieves a better balance between privacy and utility, yielding significantly higher query accuracy than the existing state-of-the-art method. In future work, we would explore combining differential privacy with other privacy-preserving mechanisms, such as homomorphic encryption and secure multi-party computation, to further strengthen privacy guarantees and provide higher security for complex data query scenarios.

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
