# A    MISSING PROOFS

## A.1    PROOF OF THEOREM 1

We restate the theorem for convenience:

**Theorem 5** (Theorem 1, restated). *Let $\mathcal{T}_H$ denote the time complexity of the hash function operations in our algorithm, and $m$ be the number of elements. The overall running time of the DLDP-BF is $O\big(m\ln 2 \cdot (\mathcal{T}_H + 1)\big)$.*

*Proof.* By combining the results of Lemma 1 and Lemma 2, we can obtain that the running time of our entire algorithm is $O(m\ln 2 \cdot (\mathcal{T}_H + 1))$. □

The Lemma 1 and Lemma 2 is proved as follows.

### A.1.1    RUNTIME FOR INSERTING PROCESS

Now, we calculate the time of the Inserting Process for our algorithm.

**Lemma 1** (Runtime of Inserting Process). *Let $\mathcal{T}_H$ denote the time to evaluate the hash function $H$ for any element $x \in S$. The Insertion Process runs in $O(m\ln 2 \cdot (\mathcal{T}_H + 1))$ time.*

*Proof.* Consider the insertion process in DLDP-BF. For each element $i \in U$, the optimal number of hash functions is:

$$h_i^* = \frac{m}{n}\ln 2 + \log_2\left(\frac{\mathcal{F}_i}{\mathcal{L}_i}\right) - \sum_{j\in U}\frac{\mathcal{L}_j}{n}\log_2\left(\frac{\mathcal{F}_j}{\mathcal{L}_j}\right)$$

A single element $i$ requires $O(h_i^* \cdot \mathcal{T}_H)$ time for hash computations. Summing over all $n = \sum_{i\in U}\mathcal{L}_i$ elements:

$$\sum_{i\in U}h_i^*\mathcal{L}_i = \frac{m}{n}\ln 2\sum_{i\in U}\mathcal{L}_i + \sum_{i\in U}\mathcal{L}_i\log_2\left(\frac{\mathcal{F}_i}{\mathcal{L}_i}\right) \quad - \sum_{i\in U}\mathcal{L}_i\left(\sum_{j\in U}\frac{\mathcal{L}_j}{n}\log_2\left(\frac{\mathcal{F}_j}{\mathcal{L}_j}\right)\right)$$

$$= m\ln 2 + \sum_{i\in U}\mathcal{L}_i\log_2\left(\frac{\mathcal{F}_i}{\mathcal{L}_i}\right) \quad - \sum_{i\in U}\mathcal{L}_i\log_2\left(\frac{\mathcal{F}_i}{\mathcal{L}_i}\right)$$

$$= m\ln 2 \quad \text{(by cancellation of cross terms)}$$

Thus, the total hashing time is $O(m\ln 2 \cdot \mathcal{T}_H)$.

For Local differential privacy noise injection, each element flips $h_i^*$ bits in $O(h_i^*)$ time. The total LDP flipping time is:

$$\sum_{i\in U}O(h_i^*\mathcal{L}_i) = O(m\ln 2)$$

Combining both steps, the total time is $O(m\ln 2 \cdot (\mathcal{T}_H + 1))$. □

### A.1.2    RUNTIME FOR QUERYING PROCESS

Then, we proceed to calculate the query time for our method.

**Lemma 2** (Runtime of Querying Process). *Let $\mathcal{T}_H$ denote the time to evaluate the hash function $H$ for any element $x \in Q$. The Querying Process runs in $O\left(\frac{m}{n}\ln 2 \cdot \mathcal{T}_H\right)$ time.*

*Proof.* We begin by considering the sum of terms involving $\mathcal{F}_i$ and $h_i^*$ for all $i \in U$:

$$\sum_{i \in U} \mathcal{F}_i \cdot h_i^* = \sum_{i \in U} \mathcal{F}_i \cdot \frac{m}{n} \ln 2 + \sum_{i \in U} \mathcal{F}_i \log_2 \left( \frac{\mathcal{F}_i}{\mathcal{L}_i} \right) \quad - \sum_{i \in U} \mathcal{F}_i \sum_{j \in U} \frac{\mathcal{L}_j}{n} \log_2 \left( \frac{\mathcal{F}_j}{\mathcal{L}_j} \right)$$

$$= \frac{m}{n} \ln 2 + \sum_{i \in U} \mathcal{F}_i \log_2 \left( \frac{\mathcal{F}_i}{\mathcal{L}_i} \right) \quad - \sum_{j \in U} \frac{\mathcal{L}_j}{n} \log_2 \left( \frac{\mathcal{F}_j}{\mathcal{L}_j} \right)$$

$$= \frac{m}{n} \ln 2 + \sum_{i \in U} \left( \mathcal{F}_i \log_2 \frac{\mathcal{F}_i}{\mathcal{L}_i/n} \right) \quad - \sum_{j \in U} \frac{\mathcal{L}_j}{n} \log_2 \frac{\mathcal{F}_j}{\mathcal{L}_j/n}.$$

At this point, we let $a_i = \mathcal{F}_i$ and $b_i = \frac{\mathcal{L}_i}{n}$, noting that $\sum_{i \in U} a_i = \sum_{i \in U} b_i = 1$. This leads us to the following expression:

$$\frac{m}{n} \ln 2 + \sum_{i \in U} a_i \log_2 \frac{a_i}{b_i} - \sum_{i \in U} b_i \log_2 \frac{a_i}{b_i}.$$

By rearranging, we obtain:

$$\frac{m}{n} \ln 2 + \sum_{i \in U} a_i \log_2 \frac{a_i}{b_i} + \sum_{i \in U} b_i \log_2 \frac{b_i}{a_i}.$$

Now, we recognize that the two terms in the sum are the Kullback-Leibler (KL) divergences. Specifically, we have:

$$\sum_{i \in U} a_i \log_2 \frac{a_i}{b_i} = KL(A \parallel B)$$

and

$$\sum_{i \in U} b_i \log_2 \frac{b_i}{a_i} = KL(B \parallel A)$$

where $A = (a_1, a_2, \ldots, a_{|U|})$ and $B = (b_1, b_2, \ldots, b_{|U|})$.

Next, we apply the inequality from the theorem in Popescu et al. (2016) (Upper bounds of relative entropy and its applications), which states that for two probability mass functions $p(x)$ and $q(x)$, the KL divergence satisfies:

$$KL(p \parallel q) \leq \sum_{x \in X} \frac{q^2(x)}{p(x)} - 1.$$

Using this inequality, we can bound the two KL divergence terms:

$$\sum_{i \in U} a_i \log_2 \frac{a_i}{b_i} \leq \sum_{i \in U} \frac{a_i^2}{b_i} - 1,$$

and similarly,

$$\sum_{i \in U} b_i \log_2 \frac{b_i}{a_i} \leq \sum_{i \in U} \frac{b_i^2}{a_i} - 1.$$

Now, let $U_{\min} = \min_{i \in U} \frac{a_i}{b_i}$ and $U_{\max} = \max_{i \in U} \frac{a_i}{b_i}$. These constants allow us to further bound the sums:

$$\sum_{i \in U} a_i \log_2 \frac{a_i}{b_i} \leq \sum_{i \in U} \frac{a_i^2}{b_i} - 1 \leq U_{\max} - 1,$$

and

$$\sum_{i \in U} b_i \log_2 \frac{b_i}{a_i} \leq \sum_{i \in U} \frac{b_i^2}{a_i} - 1 \leq \frac{1}{U_{\min}} - 1.$$

Finally, combining these bounds, we obtain:

$$\sum_{i \in U} F_i \cdot h_i^* \leq \frac{m}{n} \ln 2 + U_{\max} - 1 + \frac{1}{U_{\min}} - 1 = O \left( \frac{m}{n} \ln 2 \right),$$

where $U_{\max}$ and $U_{\min}$ are constants.

Therefore, the total time required to run the querying process for each query is $O\left(\frac{m}{n}\ln 2 \cdot \mathcal{T}_H\right)$. $\quad\square$

## A.2 PROOF OF THEOREM 2

We restate the theorem for convenience:

**Theorem 6** (Theorem 2, restated). *Let $n$ denote the total number of elements and $m$ denote the size of the Bloom Filter bit array. The overall space required by DLDP-BF is $O(m)$.*

*Proof.* The primary component of space usage in DLDP-BF is the Bloom Filter bit array, which consists of $m$ bits.

Assume that each element $i \in U$ has a membership likelihood (expected multiplicity) $\mathcal{L}_i \geq 0$ such that

$$\sum_{i \in U} \mathcal{L}_i = n.$$

The optimal number of hash functions for element $i$ is computed as

$$h_i^* = \frac{m}{n}\ln 2 + \log_2\left(\frac{\mathcal{F}_i}{\mathcal{L}_i}\right) - \sum_{j \in U}\frac{\mathcal{L}_j}{n}\log_2\left(\frac{\mathcal{F}_j}{\mathcal{L}_j}\right).$$

Each element $i$ requires $O(h_i^*)$ hash evaluations in expectation. Aggregating over all elements, the expected total number of hash evaluations is

$$\sum_{i \in U} h_i^* \mathcal{L}_i = \frac{m}{n}\ln 2 \sum_{i \in U}\mathcal{L}_i + \sum_{i \in U}\mathcal{L}_i\log_2\left(\frac{\mathcal{F}_i}{\mathcal{L}_i}\right) - \sum_{i \in U}\mathcal{L}_i\left(\sum_{j \in U}\frac{\mathcal{L}_j}{n}\log_2\left(\frac{\mathcal{F}_j}{\mathcal{L}_j}\right)\right)$$

$$= m\ln 2 + \sum_{i \in U}\mathcal{L}_i\log_2\left(\frac{\mathcal{F}_i}{\mathcal{L}_i}\right) - \sum_{i \in U}\mathcal{L}_i\log_2\left(\frac{\mathcal{F}_i}{\mathcal{L}_i}\right)$$

$$= m\ln 2.$$

In DLDP-BF, each element $i \in U$ is associated with a membership likelihood $\mathcal{L}_i$ and a normalized query frequency $\mathcal{F}_i$. These values are typically represented as floating-point numbers and require only $O(n)$ space in total for all $n$ elements. In contrast, the Bloom Filter bit array occupies $m$ bits, where $m$ is generally much larger than $n$ to achieve a low false positive rate. Therefore, the storage required for $\mathcal{L}_i$ and $\mathcal{F}_i$ is negligible compared to the Bloom Filter itself, and does not affect the overall $O(m)$ space complexity of DLDP-BF.

Therefore, the overall space complexity of DLDP-BF is

$$O(m).$$

$\square$

## A.3 PROOF OF THEOREM 3

We restate the theorem for convenience:

**Theorem 7** (Theorem 3, restated). *The Personalized Budget Allocation under Local Differential Privacy (PBA) satisfies $\epsilon_\infty$-Local differential privacy, where*

$$\epsilon_\infty = h_1^*\epsilon_1 + h_2^*\epsilon_2,$$

*and $h_1^*, h_2^*$ are the number of hash functions used for any adjacent values $v_1$ and $v_2$, respectively, while $\epsilon_1$ and $\epsilon_2$ are their corresponding privacy budgets.*

*Proof.* Let $v_1$ and $v_2$ be two adjacent values, and let $S_1$ and $S_2$ be their respective sets of the Bloom filter positions, derived from $h_1^*$ and $h_2^*$ hash functions. The maximum number of differing positions between $S_1$ and $S_2$ is $h_1^* + h_2^*$.

For each bit $B_i$ in the Bloom filter:

- If $B_i = 1$ (i.e., $i \in S_1$ or $i \in S_2$), then:

$$P(B_i' = 1 | B_i = 1) = \frac{e^{\epsilon_1}}{e^{\epsilon_1} + 1} \quad \text{or} \quad \frac{e^{\epsilon_2}}{e^{\epsilon_2} + 1},$$

and

$$P(B_i' = 0 | B_i = 1) = \frac{1}{e^{\epsilon_1} + 1} \quad \text{or} \quad \frac{1}{e^{\epsilon_2} + 1}.$$

- If $B_i = 0$ (i.e., $i \notin S_1$ and $i \notin S_2$), then:

$$P(B_i' = 1 | B_i = 0) = \frac{1}{e^{\epsilon_1} + 1} \quad \text{or} \quad \frac{1}{e^{\epsilon_2} + 1},$$

and

$$P(B_i' = 0 | B_i = 0) = \frac{e^{\epsilon_1}}{e^{\epsilon_1} + 1} \quad \text{or} \quad \frac{e^{\epsilon_2}}{e^{\epsilon_2} + 1}.$$

For any pair of adjacent values $v_1$ and $v_2$, the probability ratio of observing a perturbed Bloom filter $B'$ is bounded by the product of the probability ratios for each differing bit. Specifically:

- For $i \in S_1$ (mapped by $v_1$):

$$\frac{P(B_i' = 1 | B_i = 1)}{P(B_i' = 1 | B_i = 0)} = \frac{\frac{e^{\epsilon_1}}{e^{\epsilon_1}+1}}{\frac{1}{e^{\epsilon_1}+1}} = e^{\epsilon_1}.$$

- For $j \in S_2$ (mapped by $v_2$):

$$\frac{P(B_j' = 1 | B_j = 1)}{P(B_j' = 1 | B_j = 0)} = \frac{\frac{e^{\epsilon_2}}{e^{\epsilon_2}+1}}{\frac{1}{e^{\epsilon_2}+1}} = e^{\epsilon_2}.$$

Since there are at most $h_1^* + h_2^*$ differing positions, the overall probability ratio is:

$$\frac{P(B'|v_1)}{P(B'|v_2)} \le e^{h_1^* \epsilon_1 + h_2^* \epsilon_2}.$$

By the definition of local differential privacy, the mechanism satisfies $\epsilon_\infty$-local differential privacy, where:

$$\epsilon_\infty = h_1^* \epsilon_1 + h_2^* \epsilon_2.$$

The Permanent Randomized Response mechanism provides $\epsilon$-local differential privacy, with $\epsilon$ determined by the allocate sum of the privacy budgets $\epsilon_1$ and $\epsilon_2$ and the number of hash functions $h_1^*$ and $h_2^*$ used for adjacent values $v_1$ and $v_2$.

$$\epsilon_\infty = h_1^* \epsilon_1 + h_2^* \epsilon_2.$$

Please note that our approach provides a generalized result. In this scenario, the privacy budget deteriorates into the uniform noise case in the classic Bloom filter Erlingsson et al. (2014), which is when all elements have identical querying frequencies, the privacy budget is given by

$$\epsilon_\infty = 2h\epsilon.$$

where $h_1^* = h_2^* = h$ and $\epsilon_1 = \epsilon_2 = \epsilon$. This result indicates that when elements have identical querying frequencies and a unified privacy budget is used, the influence of the privacy budget is quantified by the above formula. □

## A.4 PROOF OF UTILITY ANALYSIS

We first present the accuracy of the query of the DHFA, as follows.

**Theorem 8** (Query Accuracy of DHFA). *For each hash function selects an array position with equal probability, The probability that $\hat{y}$ equals $y$ satisfies:*

$$\Pr[\hat{y} = y] \geq 1 - \left(1 - e^{-2\sum_{i=1}^{n} h_i^*/m}\right)^{h_i^*} \alpha.$$

*Proof.* According to the Differentiated hash function assignment algorithm (DHFA), we need to calculate the following $\Pr[\hat{y} = 1 | y = 0] = \Pr[E_1]$:

(1) Probability of a Bit Not Being Set:
   - When an element is inserted, each hash function sets a specific bit to 1 with probability $\frac{1}{m}$. Therefore, the probability that a bit remains unset after one hash function operation is $1 - \frac{1}{m}$.
   - After inserting $n$ elements, with each element using $k$ hash functions, the probability that a specific bit remains unset is:
$$\left(1 - \frac{1}{m}\right)^{\sum_{i=1}^{n} h_i^*}$$
   - Using the inequality $\left(1 - \frac{1}{m}\right)^m \geq e^{-2}$ (which holds for all $m \geq 2$), we can rewrite the above expression in exponential form:
$$\left(1 - \frac{1}{m}\right)^{\sum_{i=1}^{n} h_i^*} = \left(\left(1 - \frac{1}{m}\right)^m\right)^{\frac{\sum_{i=1}^{n} h_i^*}{m}} \geq e^{-2\sum_{i=1}^{n} h_i^*/m}$$

(2) Probability of a Bit Being Set:
   The probability that a specific bit is set to 1 after inserting $n$ elements is the complement of the previous probability, given by:
$$1 - \left(1 - \frac{1}{m}\right)^{\sum_{i=1}^{n} h_i^*}$$
   Substituting the lower bound for $\left(1 - \frac{1}{m}\right)^{\sum_{i=1}^{n} h_i^*}$, we get:
$$1 - \left(1 - \frac{1}{m}\right)^{\sum_{i=1}^{n} h_i^*} \leq 1 - e^{-2\sum_{i=1}^{n} h_i^*/m}$$

(3) Probability of False Positive Event $E_1$:

   A false positive occurs when all $k$ hash positions for a query element $y \notin S$ are set to 1. Since the hash functions are assumed to be independent, the probability of this event is the product of the probabilities of each individual bit being set to 1:
$$\Pr[E_1] = \left(1 - \left(1 - \frac{1}{m}\right)^{\sum_{i=1}^{n} h_i^*}\right)^{h_i^*}.$$
   Using the bound for the individual bit set probability:
$$\Pr[E_1] \leq \left(1 - e^{-2\sum_{i=1}^{n} h_i^*/m}\right)^{h_i^*}.$$

Let $\alpha = \Pr[y = 0]$ represent the prior probability of querying a non-member. The DHFA's overall accuracy is the probability that the filter returns the correct result, which is:
$$\Pr[\hat{y} = y] = 1 - \Pr[\hat{y} = 1 \mid y = 0] \cdot \alpha = 1 - \Pr[E_1] \cdot \alpha.$$
Substituting the upper bound for $\Pr[E_1]$, we obtain the final bound for the DHFA's accuracy:
$$\Pr[\hat{y} = y] \geq 1 - \alpha \left(1 - e^{-2\sum_{i=1}^{n} h_i^*/m}\right)^{h_i^*}.$$

$\square$

We then assess the accuracy loss caused by the introduction of the random response technique by comparing the outputs of the DLDP-BF with those of the DHFA.

**Theorem 9** (Query Accuracy of DLDP-BF and DHFA). *For each hash function that selects an array position with equal probability, the probability that $\hat{y}$ equals $\tilde{y}$ satisfies:*

$$\Pr[\tilde{y} = \hat{y}] \geq t \cdot \alpha \cdot \left( 1 - \left( 1 - e^{-2 \sum_{i=1}^n h_i^*/m} \right)^{h_i^*} \right),$$

*where $t$ and $\alpha$ are as defined below.*

*Proof.* Let $Q[i]$ represent the $i$-th element in the query set $Q$.

For the hash function $h_i^*$, we apply basic probability rules to compute the desired probability.

$$\Pr[\tilde{y} = \hat{y}] = \Pr[\tilde{y} = 1|\hat{y} = 1] \cdot \Pr[\hat{y} = 1] + \Pr[\tilde{y} = 0|\hat{y} = 0] \cdot \Pr[\hat{y} = 0].$$

Step 1: Calculate $\Pr[\tilde{y} = 1|\hat{y} = 1]$

Define the event $E_2$ as follows:
$$E_2 : \tilde{g}[j] = g[j], \quad \forall j \in Q.$$
By the definition of the Bloom filter, we have:
$$\Pr[\tilde{y} = 1|\hat{y} = 1] = \Pr[E_2].$$
Now, we compute the probability that $E_2$ occurs:
$$\Pr[E_2] = \prod_{i=1}^{k} \Pr[\tilde{g}[Q[i]] = g[Q[i]]] = \left( \frac{e^{\epsilon_i^*}}{e^{\epsilon_i^*} + 1} \right)^{h_i^*}.$$
Thus, we obtain the following:
$$\Pr[\tilde{y} = 1|\hat{y} = 1] = \left( \frac{e^{\epsilon_i^*}}{e^{\epsilon_i^*} + 1} \right)^{h_i^*}.$$

Step 2: Calculate $\Pr[\tilde{y} = 0|\hat{y} = 0]$

Let $Q$ be a subset of $[m]$, and define the set $Z$ as the collection of elements in $Q$ where $g[j] = 0$.

For the $i$-th element of $Z$, denote it as $Z[i]$. Additionally, define the complementary set $\overline{Q}$ as the set obtained by removing the elements of $Z$ from $Q$.

By basic probability rules, we have:
$$\Pr[\tilde{y} = 0|\hat{y} = 0] = 1 - \Pr[\tilde{y} = 1|\hat{y} = 0].$$
Next, we calculate $\Pr[\tilde{y} = 1|\hat{y} = 0]$. This event occurs if and only if the following conditions are met:

- All elements in $Z$ flip from 0 to 1.

- All elements in $\overline{Q}$ remain 1.

Thus, we have:

$$\Pr[\tilde{y} = 1|\hat{y} = 0] = \prod_{i=1}^{|Z|} \Pr[\tilde{g}[Z[i]] = 1] \cdot \prod_{i=1}^{|\overline{Q}|} \Pr[\tilde{g}[\overline{Q}[i]] = 1]$$

$$= \left( \frac{1}{e^{\epsilon_i^*} + 1} \right)^{|Z|} \cdot \left( \frac{e^{\epsilon_i^*}}{e^{\epsilon_i^*} + 1} \right)^{|\overline{Q}|}$$

$$\leq \left( \frac{1}{e^{\epsilon_i^*} + 1} \right)^{|Z|}$$

$$\leq \frac{1}{e^{\epsilon_i^*} + 1}.$$

Then, we obtain:

$$\Pr[\tilde{y} = 0|\hat{y} = 0] = 1 - \Pr[\tilde{y} = 1|\hat{y} = 0] \geq 1 - \frac{1}{e^{\epsilon_i^*} + 1} = \frac{e^{\epsilon_i^*}}{e^{\epsilon_i^*} + 1}.$$

Let $\hat{\alpha} := \Pr[\tilde{y} = 0]$, so that $1 - \hat{\alpha} = \Pr[\tilde{y} = 1]$. Also, define $\alpha := \Pr[y = 0]$. Note that:

$$\hat{\alpha} = \alpha \left( 1 - \left( 1 - e^{-2\sum_{i=1}^{n} h_i^*/m} \right)^{h_i^*} \right).$$

Let $t := \frac{e^{\epsilon_i^*}}{e^{\epsilon_i^*} + 1}$.

The final query accuracy is given by:

$$\Pr[\tilde{y} = 0|\hat{y} = 0] \cdot \Pr[\hat{y} = 0] + \Pr[\tilde{y} = 1|\hat{y} = 1] \cdot \Pr[\hat{y} = 1]$$

$$= \Pr[\tilde{y} = 0|\hat{y} = 0] \cdot \hat{\alpha} + \Pr[\tilde{y} = 1|\hat{y} = 1] \cdot (1 - \hat{\alpha})$$

$$= \Pr[\tilde{y} = 0|\hat{y} = 0] \cdot \alpha (1 - \left( 1 - e^{-2\sum_{i=1}^{n} h_i^*/m} \right)^{h_i^*})$$

$$+ \Pr[\tilde{y} = 1|\hat{y} = 1] \cdot (1 - \alpha + \alpha \cdot \left( 1 - e^{-2\sum_{i=1}^{n} h_i^*/m} \right)^{h_i^*})$$

$$\geq \frac{e^{\epsilon_i^*}}{e^{\epsilon_i^*} + 1} \cdot \alpha (1 - \left( 1 - e^{-2\sum_{i=1}^{n} h_i^*/m} \right)^{h_i^*})$$

$$+ \left( \frac{e^{\epsilon_i^*}}{e^{\epsilon_i^*} + 1} \right)^{h_i^*} \cdot (1 - \alpha + \alpha \cdot \left( 1 - e^{-2\sum_{i=1}^{n} h_i^*/m} \right)^{h_i^*})$$

$$= t \cdot \alpha - t \cdot \alpha \cdot \left( 1 - e^{-2\sum_{i=1}^{n} h_i^*/m} \right)^{h_i^*} + t^{h_i^*} \cdot (1 - \alpha + \alpha \cdot \left( 1 - e^{-2\sum_{i=1}^{n} h_i^*/m} \right)^{h_i^*})$$

$$\geq t \cdot \alpha \cdot (1 - \left( 1 - e^{-2\sum_{i=1}^{n} h_i^*/m} \right)^{h_i^*}).$$

$\square$

Now, we can examine the utility guarantees of DLDP-BF by calculating the error between the ground truth for the query and the output of DLDP-BF.

**Theorem 10** (Theorem 4, restated). *The probability that $\tilde{y}$ equals $y$ satisfies:*

$$\Pr[\tilde{y} = y] \geq \alpha(1 - t - t^{h_i^*}) \left( 1 - e^{-2\sum_{i=1}^{n} h_i^*/m} \right)^{h_i^*} + \alpha t.$$

*Proof.*

$$\Pr[\tilde{y} = y]$$
$$= \Pr[\tilde{y} = 0|\hat{y} = 0] \Pr[\hat{y} = 0|y = 0] \Pr[y = 0]$$
$$+ \Pr[\tilde{y} = 0|\hat{y} = 1] \Pr[\hat{y} = 1|y = 0] \Pr[y = 0]$$
$$+ \Pr[\tilde{y} = 1|\hat{y} = 1] \Pr[\hat{y} = 1|y = 1] \Pr[y = 1]$$
$$+ \Pr[\tilde{y} = 1|\hat{y} = 0] \Pr[\hat{y} = 0|y = 1] \Pr[y = 1]$$
$$\geq t \cdot (1 - \Pr[E_1]) \cdot \alpha + (1 - t^{h_i^*}) \cdot \Pr[E_1] \cdot \alpha + t^{h_i^*} \cdot 1 \cdot (1 - \alpha)$$
$$\geq \alpha(1 - t - t^{h_i^*}) \left( 1 - e^{-2\sum_{i=1}^{n} h_i^*/m} \right)^{h_i^*} + \alpha t + t^{h_i^*}(1 - \alpha)$$
$$\geq \alpha(1 - t - t^{h_i^*}) \left( 1 - e^{-2\sum_{i=1}^{n} h_i^*/m} \right)^{h_i^*} + \alpha t.$$

$\square$

# B MORE DETAILS ON EXPERIMENTAL SETTINGS

## B.1 DATASET USED IN THE EXPERIMENTS

We test the proposed methods on five real datasets.

- **Taxi** The Taxi dataset consists of yellow taxi trip records from New York City, spanning January to December 2019. It includes details such as pick-up and drop-off times, locations, trip distances, fares, and more. The dataset contains approximately 77 million taxi trips. For each trip, we simulated client data and applied it to the sliding window for real-time processing. The yellow taxi trip dataset is available at https://catalog.data.gov/dataset/2019-yellow-taxi-trip-data.

- **Accident** The Accident dataset is a detailed road safety dataset about personal injury. Road accidents are reported by the British police, who collect data on every car collision. The dataset contains information such as age and gender from the casualty files from 2005 to 2014 was selected as the experimental dataset. The UK Car Accident Dataset is available at https://github.com/cbeludru/Road-Accident-Analysis.

- **Obesity** The Obesity Level Prediction Dataset is designed to predict obesity levels based on various factors such as age, gender, lifestyle habits, and family history of overweight. The dataset contains information on 2111 individuals and their obesity levels, ranging from normal weight to type III obesity. The Obesity Level Prediction Dataset is available at https://github.com/shaecodes/Obesity-Level-Prediction.

- **Letter Recognition** The Letter Recognition dataset aims to classify capital letters of the English alphabet based on attributes extracted from images. It contains 20,000 examples of handwritten uppercase letters (A–Z), where each instance is represented by 16 numerical features derived from pixel-based statistical descriptors such as width, height, edge counts, and symmetry. The goal of this dataset is to map these extracted features to one of the 26 alphabet classes for pattern recognition tasks. The Letter Recognition dataset is available at https://archive.ics.uci.edu/dataset/59/letter+recognition.

- **ISOLET** The ISOLET dataset is designed for speech recognition of isolated English letters (A–Z). It contains recordings from 150 speakers, with each speaker pronouncing each letter twice, resulting in 7,797 instances. Each instance is represented by 617 numerical features extracted from audio signals, capturing characteristics such as frequency, spectral coefficients, and temporal dynamics. The task is to classify each instance into one of the 26 letter classes based on the audio-derived features. The ISOLET dataset is available at https://archive.ics.uci.edu/dataset/54/isolet.

### B.2 MORE BASELINES IN ADDITION TO SECTION 6

For better comparisons, these methods are considered in the experiments.

- **Non_Privacy** Bloom (1970): The Non_Privacy method serves as a baseline, where a Bloom Filter uses a fixed-length bit array and multiple independent hash functions to map elements to specific positions and set them to 1, without introducing any noise into the bit array. This method relies entirely on deterministic insertion and query rules to perform membership tests, but it provides no privacy protection.

- **RAPPOR** Erlingsson et al. (2014): RAPPOR applies a two-stage randomized response to protect membership queries under strong local privacy. In the first stage, known as permanent randomized response, each bit in the Bloom filter representation of the input is perturbed once and stored persistently, creating a privacy-preserving encoding of the client data. In the second stage, called instantaneous randomized response, the stored bits are further perturbed at each query instance before being used for membership testing, ensuring that repeated queries do not leak information about the original data.

- **DPBloomFilter** Ke et al. (2025): DPBloomFilter directly injects differential privacy noise into the Bloom filter bit array in a single step. Each bit is perturbed according to a calibrated noise distribution, ensuring that membership queries over the Bloom filter satisfy strict local differential privacy while maintaining reasonable query accuracy.

- **DLDP-BF (Proposed)**: We propose A Differentiated Local Differential Privacy Bloom Filter for Membership Queries (DLDP-BF). It dynamically allocates hash functions and privacy budgets according to the importance of data, aiming to achieve high utility under rigorous local differential privacy guarantees.

B.3 MEASUREMENT METRICS

Root Mean Square Error (RMSE) Erlingsson et al. (2014); Ghazi et al. (2021); Vatsalan et al. (2022) quantifies the average deviation between the true query results and the outputs produced by the differentially private mechanisms. It is formally defined as:

$$\text{RMSE} = \sqrt{\frac{1}{N} \sum_{i=1}^{N} \left( \psi_i - \hat{\psi}_i \right)^2}$$

where $\psi_i$ denotes the true membership status of the $i$-th element, $\hat{\psi}_i$ is the corresponding predicted result, and $N$ is the total number of queries. A lower RMSE indicates that the mechanism produces results closer to the ground truth, thereby reflecting higher utility. Notably, there exists an inherent trade-off between privacy and utility: a smaller privacy budget $\epsilon$ ensures stronger privacy protection but typically leads to higher RMSE due to increased noise; conversely, a larger $\epsilon$ allows more accurate outputs but provides weaker privacy guarantees.

In addition to RMSE, we also report the Accuracy of the predicted results to evaluate the correctness of membership inference Song & Mittal (2021); Torre et al. (2025). Accuracy is defined as the proportion of correctly predicted outcomes over all queries:

$$\text{Accuracy} = \frac{1}{N} \sum_{i=1}^{N} \mathbb{I} \left[ \psi_i = \hat{\psi}_i \right]$$

where $\mathbb{I}[\cdot]$ is the indicator function, which equals 1 when the condition holds and 0 otherwise. Higher accuracy indicates better consistency with the true membership states. Similar to RMSE, Accuracy is influenced by the privacy budget, often exhibiting a positive correlation with $\epsilon$.

## C MORE EXPERIMENTAL RESULTS

### C.1 DETAILED ANALYSIS OF SECTION 6.2.1

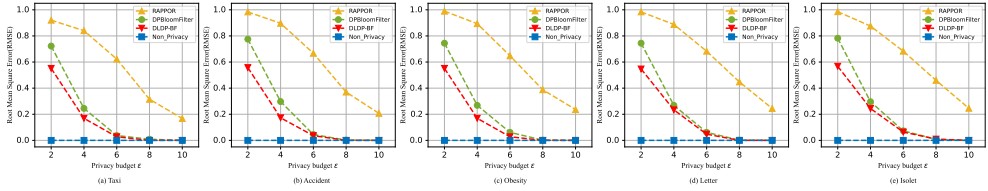

Figure 7: The RMSE of differential privacy budget $\epsilon$

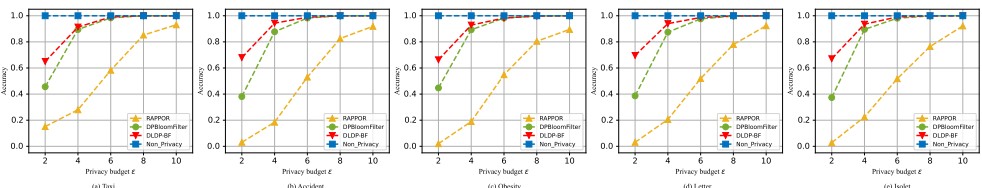

Figure 8: The Accuracy of differential privacy budget $\epsilon$

We analyze the impact of different privacy budget values ($\epsilon$) on RMSE and accuracy. As shown in Fig. 7 and Fig. 8, as $\epsilon$ increases, RMSE exhibits an overall downward trend, while accuracy steadily improves, reflecting that looser privacy constraints enhance data utility.

Taking the *Non_Privacy* method as the baseline, DLDP-BF achieves performance that is close to the utility upper bound, while still satisfying local differential privacy. Compared with RAPPOR and

DPBloomFilter, DLDP-BF achieves better noise control and maintains a smaller RMSE and higher accuracy. This improvement stems from the joint use of the differentiated hash function assignment and the personalized budget allocation under local differential privacy, which together enhance data usability.

Overall, relative to the best existing differential privacy methods, DLDP-BF achieves a **49.0% reduction in RMSE** and a **12.3% improvement in accuracy**, while remaining much closer to the *Non_Privacy* baseline. These results highlight the effectiveness of DLDP-BF in achieving a favorable privacy–utility balance across different privacy budgets.

## C.2 DETAILED ANALYSIS OF SECTION 6.2.2

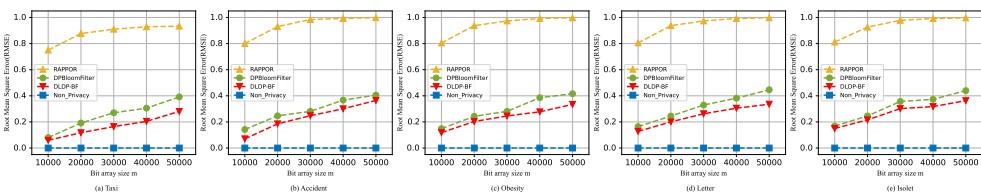

Figure 9: The RMSE of differential Bit array size $m$

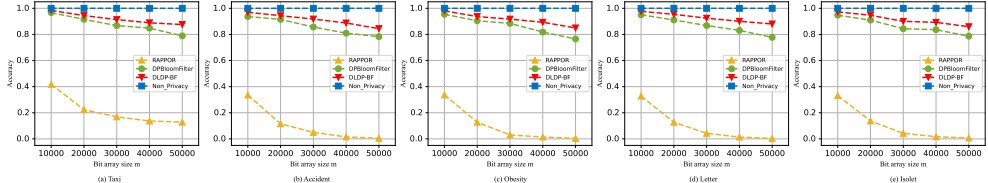

Figure 10: The Accuracy of differential Bit array size $m$

We analyze the impact of different bit array sizes ($m$) on RMSE and accuracy. As shown in Fig. 9 and Fig.10, as $m$ increases, RMSE shows an overall upward trend, while accuracy decreases, reflecting that larger bit array sizes reduce data utility.

Taking the *Non_Privacy* method as the baseline, DLDP-BF maintains performance close to the utility upper bound while still providing rigorous local differential privacy guarantees. Compared with RAPPOR and DPBloomFilter, DLDP-BF achieves better noise control and consistently yields smaller RMSE and higher accuracy. This advantage stems from the integration of the differentiated hash function assignment algorithm with the personalized budget allocation strategy, which together enhance data usability.

Overall, relative to the best existing differential privacy methods, DLDP-BF achieves a **25.2% reduction in RMSE** and a **5.8% improvement in accuracy**, while remaining much closer to the *Non_Privacy* baseline. These results demonstrate that DLDP-BF can effectively maintain data usability under different bit array sizes, highlighting its strength in balancing privacy and utility.