# OpenReview forum: "DLDP-BF: A Differentiated Local Differential Privacy Bloom Filter for Membership Queries"
_ICLR.cc/2026/Conference — Submitted to ICLR 2026_

### Official Review · Reviewer_wb3W · 2025-10-29

**Soundness:** 3
**Presentation:** 2
**Contribution:** 2
**Rating:** 4
**Confidence:** 2

**Summary:**

This paper proposes DLDP-BF, a Differentiated Local Differential Privacy Bloom Filter that addresses limitations in existing LDP Bloom filter methods by dynamically allocating hash functions and privacy budgets based on element importance. The authors introduce two algorithms: DHFA (Differentiated Hash Function Assignment) that assigns more hash functions to frequently queried elements, and PBA (Personalized Budget Allocation) that allocates privacy budgets proportionally to query frequency and membership probability. Experiments on three datasets demonstrate 49.0% RMSE reduction and 12.3% accuracy improvement compared to RAPPOR and DPBloomFilter, though the evaluation scope remains limited.

**Strengths:**

1. Well-motivated technical contribution with theoretical support. The paper identifies two genuine limitations in existing work: uniform hash function assignment regardless of data importance and fixed privacy budget allocation across all elements (lines 086-098). The proposed differentiated approach is intuitive and addresses a real practical need. The theoretical analysis (Section 4) provides formal privacy guarantees (Theorem 2) and utility bounds (Theorem 3), with the privacy proof properly accounting for the permanent randomized response mechanism.


2. New privacy budget allocation framework. The PBA algorithm (Algorithm 3, Equation 2) represents, to the authors' knowledge, the first local differential privacy budget allocation method that jointly considers membership probability and query frequency (lines 106-107). This personalized approach is more realistic than uniform allocation and the formulation elegantly balances privacy protection for critical elements while maintaining utility for frequently queried items.

**Weaknesses:**

1. Insufficient experimental evaluation. The experiments only compare against three methods (Non_Privacy, RAPPOR, DPBloomFilter) on three datasets (Section 5.1), which is limited for demonstrating broad applicability. Critically, the paper claims DPBloomFilter (Ke et al. 2025) is "a representative solution" (line 068) yet this is a very recent concurrent work that may not represent the state-of-the-art.

2. Critical methodological details are unclear. The paper does not explain how query frequency Fi and membership likelihood Li are obtained in practice. Are these estimated from historical data, provided by applications, or learned? This is crucial since the entire method depends on these values (Algorithm 2, line 258; Algorithm 3, line 272). The DHFA algorithm (Equation 1) computes optimal hash count h*i but line 268 states "Let the expected size of the set be n" without clarifying whether n is known beforehand or estimated.

3. Privacy-utility trade-off analysis lacks depth and the privacy model has limitations. While Figure 3 and 4 show performance across different ε values, the paper does not analyze the fundamental trade-off: how much privacy is sacrificed for utility gains compared to uniform allocation? For instance, does allocating more budget to high-frequency elements disproportionately expose their membership? The privacy guarantee (Theorem 2) assumes the hash function family and number of hash functions are public (Definition 2, lines 146-149), but the differentiated assignment itself may leak information about element importance.

**Questions:**

1. Why was DPBloomFilter chosen as the primary comparison method rather than earlier established LDP Bloom filter works?

2. What happens in cold-start scenarios where no historical data exists for new elements?

3. For membership likelihood, how do you estimate the probability that an element belongs to a set before actually querying it?

---

> ### Author Response · Authors · 2025-11-27
>
> W1: To strengthen the persuasiveness and breadth of our empirical study, the revised manuscript now includes additional experiments on two more datasets, thereby providing a more comprehensive assessment of the applicability and robustness of our method. Furthermore, we have carefully revised our description of \textit{DPBloomFilter}. Instead of referring to it as “a representative solution,” we now explicitly characterize it as “one of the most recent solutions” to clarify that it is a concurrent, up-to-date baseline rather than a definitive state-of-the-art method. This modification ensures an accurate and fair contextualization of related work and avoids overstating the role of \textit{DPBloomFilter} in the comparison.
>
> W2: In the revised manuscript, we have added a detailed explanation of how query frequency and membership likelihood are obtained. Specifically, we clarify that the required query frequency $F_i$ and membership likelihood $L_i$ are derived from prior knowledge, consistent with existing works that leverage historical or application-level statistics to guide parameter settings, such as (Jinyuan Jia. INFOCOM 2019.), (Terrance Liu. PMLR 2021.), and (Fei Wei. VLDB 2024.). In practical deployments, these values can be sourced from historical workload traces, application semantics, or lightweight online profiling when only partial information is available. Once $F_i$ and $L_i$ are determined, the set size $n$ in the DHFA algorithm—representing the size of the target dataset—can be directly used to compute the optimal number of hash functions, ensuring that DHFA operates with well-defined and practically obtainable inputs.
>
> W3:  Our focus is on optimizing data utility under a fixed privacy budget, achieved by allocating the budget differentially while strictly preserving privacy. Following the LDP principle, the server only observes perturbed query outputs and has no access to internal states; thus, assigning a larger budget to high-frequency elements does not reveal their membership or significance. The number of hash functions per element is similarly perturbed to prevent additional leakage. As a result, the privacy-utility trade-off is explicitly controlled in our method: utility gains from non-uniform allocation improve accuracy without violating the LDP guarantee (Theorem 3), even though the hash function family and counts are public, and the differentiated assignment does not expose meaningful information about element importance, ensuring that privacy loss is minimal relative to the observed utility improvements.
>
> Q1: Regarding membership queries using Bloom filters under Local-Differential-Privacy (LDP), there are relatively few publicly available and citationable works. While we reviewed some relevant literature during our research, most studies differed significantly from our work in terms of application scenarios, assumptions, or experimental settings, making direct comparison impossible. Therefore, we chose \textit{DPBloomFilter} as our primary comparison method because it is closest to our method in terms of problem setting and evaluation metrics, providing a fair and meaningful comparison.
>
> Q2: In the cold start scenario, since new elements lack historical statistical information or category characteristics, we adopt a unified initialization strategy: assign a default number of hash functions and a default privacy budget to them based on the preset importance function.
>
> Q3: In the proposed method, the membership likelihood of an element is derived from historical statistics within the system. Before the actual query, its membership likelihood can be estimated by counting the number of times the element appeared in past sets and dividing by the total number of times in the sets.

---

### Official Review · Reviewer_Q7gb · 2025-10-29

**Soundness:** 2
**Presentation:** 2
**Contribution:** 3
**Rating:** 4
**Confidence:** 2

**Summary:**

This paper studies privacy-preserving membership queries with Bloom filters under LDP. It argues that existing LDP-BF methods use uniform parameters (same number of hash functions, same privacy budget) for all elements, which can be inefficient when elements differ in importance (e.g., membership likelihood or query frequency). The authors propose DLDP-BF, which differentiates both (i) the number of hash functions and (ii) the local privacy budget per element according to importance, aiming to improve utility at fixed privacy.

**Strengths:**

S1: The problem studied in this paper is clearly defined and well-motivated.

S2: The experimental results demonstrate performance gains, showing that the proposed approach outperforms four existing baseline methods across multiple datasets.

S3: The paper provides theoretical analysis covering computational complexity, privacy guarantees, and utility bounds.

**Weaknesses:**

W1: More clarification of the threat model is expected. The method assumes access to (or accurate estimation of) per-element membership likelihood and query frequency to drive DHFA/PBA, but it does not specify how to obtain these safely under privacy constraints or how robust the system is to estimation errors or distribution shift.

W2: Since the proposed method assigns a larger $h_i$ to high-importance items, a server can possibly infer an item’s importance class from the number/structure of touched bits. It is better to analyze or bound this ‘importance-label leakage,’ or propose a mitigation.

W3: Similarly, since the proposed method assigns larger privacy budgets to high-importance elements, a server may be able to infer an item’s importance level based on differences in noise magnitude or output variance.

W4: The DHFA logic leverages prior knowledge of frequency/likelihood distributions; however, the paper does not analyze worst-case mismatch.

W5: Experiments report RMSE and accuracy across datasets and parameters, but there is no measurement of runtime/throughput, memory, or client/server communication overhead, which are important for Bloom-filter pipelines at scale.

W6: Algorithms 1–3 need clearer, step-by-step explanations.

**Questions:**

Q1: Could the authors clarify the assumed threat model in more detail? Specifically, how are per-element membership likelihood and query frequency obtained or estimated without violating privacy guarantees?

Q2: How robust is the proposed framework if these statistics are inaccurate or shift over time?

Q3: Since the proposed method assigns a larger privacy budget/a larger number of hash functions (h_i) to higher-importance items, could a server or an attacker potentially infer an element’s importance level or class from observable Bloom filter updates?

Q4: Similarly, as the proposed method allocates larger privacy budgets to high-importance elements, could the server infer an item's importance level from the differing noise magnitude or output variance?

---

> ### Author Response · Authors · 2025-11-27
>
> W1: We have clearly reiterated the threat model. Under the assumption that the DP attacker possesses maximum background data knowledge, user membership probabilities and query frequencies do not reveal any internal information if the attacker does not know the method's parameters.
>
> W2: The proposed method locally perturbs all elements at the user end to satisfy $\epsilon$-LDP. The server can only see the perturbed member query results and cannot access the internal state of the Bloom Filter. Therefore, even if high-importance elements are allocated a larger privacy budget or use a different number of hash functions, their importance information is randomized and masked, preventing additional "importance label leakage."
>
> W3: The proposed method locally perturbs all elements at the user end to satisfy $\epsilon$-LDP, and the server only sees the perturbed query output. Even if high-importance elements are allocated a larger privacy budget, the randomness and perturbation of the output still mask the noise amplitude and output variance, thus preventing the server from inferring the importance of elements and avoiding additional privacy leakage.
>
> W4: We follow the assumptions adopted in prior work, including (Jinyuan Jia. INFOCOM 2019) and (Fei Wei. VLDB 2024), which leverage prior knowledge to guide parameter configuration and consider the frequency or likelihood distributions to be reasonably reliable and stable. Under this assumption, severe worst-case mismatches are unlikely to occur during the configuration of hash functions and privacy budgets.
>
> W5: Since this work primarily focuses on improving data availability within the same privacy budget, the initial draft did not include metrics related to algorithmic overhead. Furthermore, similar studies in the existing literature typically do not use runtime or communication overhead as primary evaluation metrics. To make the evaluation more comprehensive, we have added relevant theoretical analysis of overhead in the revised draft to more systematically demonstrate the method's performance in terms of time and space complexity.
>
> W6: We have provided step-by-step pseudocode with line-by-line explanations for Algorithms 1–3 in the revised version.
>
> Q1: The DLDP-BF framework addresses uncertainty in external inputs through its inherent algorithmic design, thus exhibiting significant robustness even with statistical errors. Its tolerance to estimation bias stems primarily from the core mechanism's reliance on the relative structure of importance among elements rather than precise absolute values. When distribution shifts, a possible solution is to introduce a periodic parameter recalibration mechanism, maintaining method utility while upholding differential privacy guarantees.
>
> Q2: The DLDP-BF framework exhibits significant robustness by addressing the uncertainty of external inputs through its inherent algorithmic design. Its tolerance to statistical errors stems primarily from the fact that its core mechanism relies on the relative order of importance among elements rather than absolute precision. To address distribution shifts, a possible solution is to introduce a periodic parameter recalibration mechanism to maintain the method's utility while preserving differential privacy.
>
> Q3: Although this method allocates a larger privacy budget or more hash functions ($h_i$) to highly important elements, the core guarantee of differential privacy still holds: the impact of a single user's update to the Bloom filter is strictly limited, so attackers cannot directly infer the specific importance or category of an element from the output or update of the Bloom filter.
>
> Q4: Although the method allocates a larger privacy budget to highly important elements, the core guarantee of differential privacy still holds: the influence of a single user on the output is strictly limited. Therefore, even if the noise amplitude or variance of the output varies among different elements, an attacker cannot accurately infer the information of a single user or the specific importance of an element.

---

### Official Review · Reviewer_sE9T · 2025-10-31

**Soundness:** 3
**Presentation:** 3
**Contribution:** 2
**Rating:** 4
**Confidence:** 3

**Summary:**

This paper investigates the problem of privacy-preserving membership queries over large-scale datasets. Traditional Bloom filter-based approaches typically employ a fixed number of hash functions for all elements, overlooking differences in their importance or frequency, which leads to a suboptimal balance between privacy and utility.

The authors propose a Differentiated Local Differential Privacy Bloom Filter (DLDP-BF), which dynamically allocates the number of hash functions and privacy budgets according to element importance, thereby improving query accuracy. The paper also designs a novel LDP budget allocation algorithm that adaptively adjusts noise intensity proportionate to element importance. Theoretical analysis proves that the method provides strict Local Differential Privacy (LDP) guarantees while enhancing data utility. Experimental results demonstrate the superior performance of the proposed approach in real-world scenarios, confirming its practical value.

**Strengths:**

1.  **Originality**: This paper innovatively incorporates element importance into privacy budgeting and hash function allocation to achieve differential privacy protection. The introduction of Personalized Budget Allocation (PBA) represents a novel direction in privacy-preserving membership query scenarios.

2.  **Clarity**: Figure 2 provides an intuitive illustration of the DLDP-BF workflow, effectively facilitating the understanding of inter-modular relationships within the system. The paper features a complete structure, with precise textual expression and coherent logical flow.

**Weaknesses:**

1. Using Bloom filter with LDP for privacy-preserving membership query is not well-motivated. The competitor RAPPOR is originally designed for statistical frequency queries.

2. The proposed method requires prior knowledge about importance/frequency of elements, which can be hard obtain in privacy-sensitive scenarios.

3. While the paper provides proofs for privacy guarantees and complexity analysis, it does not establish theoretical performance bounds under varying importance distributions .

4.  The adaptability of DLDP-BF requires further validation in dynamic query scenarios or under distribution drift conditions.

**Questions:**

1. Please provide motivation scenarios of Bloom filter with LDP for privacy-preserving membership query.

2. Please discuss the practicalness of requiring prior statistical profile about elements.

3. Please discuss the permances under distribution drift.

---

> ### Author Response · Authors · 2025-11-27
>
> W1: Although RAPPOR was originally used for frequency estimation, employing a bloom filter and perturbations with local differential privacy, it can also be applied to membership query tasks. We use it as a benchmark to evaluate its performance in membership query scenarios and compare it with our method.
>
> W2: Regarding the concern that the method requires prior knowledge about the importance or frequency of elements, existing studies in this area commonly rely on similar assumptions. Many prior works  (Jinyuan Jia. INFOCOM 2019.) (Terrance Liu. PMLR 2021.) and (Fei Wei. VLDB 2024.) assume access to certain statistical priors or auxiliary information to guide parameter configuration or mechanism design. Therefore, relying on such priors is not a limitation unique to our method but rather a common setting in this research direction.
>
> W3: For any distribution, our method has considered the relevant theoretical guarantees under different importance distributions. Even if the distributions differ significantly, the theoretical guarantees can still be guaranteed.
>
> W4: For dynamic queries, the distribution may change, while our method assumes that the distribution remains approximately stable. We plan to explore this further in future research. Regarding parameter configuration, our use of prior knowledge is consistent with existing research, such as (Jinyuan Jia, INFOCOM 2019), (Terrance Liu, PMLR 2021), and (Fei Wei, VLDB 2024); these methods also rely on the assumption that the distribution is substantially constant.
>
> Q1: LDP Bloom Filters are suitable for scenarios requiring member testing without exposing the query content, such as malicious URL detection, cross-platform account deduplication, and distributed blacklist queries. In these tasks, the client can securely send queries to the server simply by perturbing the Bloom Filter encoding locally, satisfying $\epsilon$-LDP while maintaining low communication and computational overhead.
>
> Q2: Requiring prior statistical information about elements is common in mechanisms that allocate privacy budgets or rely on element importance.
> Indeed, many current differential privacy methods use such prior knowledge to guide parameter configuration.
> For example, (Jinyuan Jia. INFOCOM 2019.) explicitly treat the true frequency distribution as a priori; (Fei Wei. VLDB 2024.) guides perturbations by estimating the data descriptor. Our method, also under this common assumption, optimizes the hash function and privacy budget allocation through prior statistics to improve utility while maintaining $\epsilon$-LDP privacy guarantees.
>
> Q3: Our method assumes that the distribution of the target set remains approximately stable, i.e., there is no significant distribution drift. This stability assumption is consistent with previous studies that rely on prior knowledge, such as (Jinyuan Jia, INFOCOM 2019), (Terrance Liu, PMLR 2021), and (Fei Wei, VLDB 2024). If the distribution changes significantly, a possible solution is through a periodic parameter recalibration mechanism.

---

### Official Review · Reviewer_kZsJ · 2025-11-04

**Soundness:** 1
**Presentation:** 2
**Contribution:** 1
**Rating:** 2
**Confidence:** 4

**Summary:**

This paper proposes a novel variant of locally differentially-private Bloom filter in which the number of hash functions assigned as well as the privacy budget allocated are chosen in a data-dependent manner to improve on the utility.

**Strengths:**

-The authors have clearly introduced the background on Bloom filters, including their efficiency for performing membership queries but also the privacy issues associated to their uses.

-The proposed approach has been validated experimentally on three different datasets and compare to two other local differentially-private versions of Bloom filters and seem to display a higher performance in terms of utility.

**Weaknesses:**

-The review of local differentially-private Bloom filters methods should be strengthen and detailed more. For instance, currently only two methods are cited but without providing any details on the underlying constructions. This issue is really important to be able to position the approach proposed and assess its novelty. Additionally, the literature review should also refer to papers that have studied how to integrate heterogenous or personalized privacy budget in a differentially-private mechanism.

-The dynamic adjustment of the hash function and the privacy budget requires to be able to store a significant amount of information on each element, which defeats the purpose of the Bloom filter that requires to have a highly efficient data structure that can be updated on the fly. More precisely, the choice of the hyper parameters of the proposed data structure should depend on a set of realistic assumptions about the distribution but not require to store an amount of data that is significantly larger than the size of the data structure.

-The dynamic adjustment of the hash function assignment and the privacy budget based on an element importance is directly in tension with privacy as estimating one of these two parameters can directly leak information about the element. This aspect is crucial to the privacy analysis but is not discussed at all in the paper, which raises serious doubts about the privacy protection provided.

-The papers are not appropriately cited in the text. In particular, there are not between parentheses.

Minor typo :
-« the overall privacy security of the system »

**Questions:**

Please see the main points raised in the weaknesses section.

---

> ### Author Response · Authors · 2025-11-27
>
> Q1: We have supplemented the revised manuscript with a more complete review of related work on locally differential privacy Bloom Filters and detail the core mechanisms of existing methods to better position the contribution of this paper. In addition, we have supplemented the literature on heterogeneous or personalized privacy budget allocation to enrich the background. This can more clearly demonstrate the innovativeness and technical position of our approach.
>
> Q2: Our method does not defeat the purpose of the Bloom filter. Instead, assuming prior knowledge to optimise Bloom filters has been widely adopted by many prior works (Jinyuan Jia. INFOCOM 2019.) (Terrance Liu. PMLR 2021.)  (Fei Wei. VLDB 2024.) to further improve the performance of Bloom filters. We follow those state-of-the-art works to further improve the bloom filter in differential privacy scenarios.
>
> Q3: The DP model we consider focuses on privacy protection at the output level, and does not involve the privacy of the model parameters themselves (i.e., model parameter privacy), which is a separate research issue. In the differential privacy model set by Dwork et al. (Dwork. ICALP 2006.), the server cannot access the internal parameters, so this parameter allocation itself does not introduce additional privacy risks.
>
> Q4: We have checked and revised the citation format in the text to ensure that all references are marked with the correct parentheses.
>
> Q5: We have corrected the spelling in that section and checked the entire text to ensure there are no similar errors.

---

### Comment · Area_Chair_p4vG · 2025-11-28
**A gentle reminder to participate in the author–reviewer discussion.**

Dear Reviewers,

Thank you once again for your service to ICLR 2026. Now that the authors have submitted their rebuttal, could you please engage in the interactive discussion with them? Your participation would be very helpful to the authors, and they would greatly appreciate it. Please also read the authors’ response together with the other reviews and consider whether the rebuttal or any additional comments influence your assessment of the paper.

Thank you again for your efforts.

Best wishes,

Your AC

---

> ### Author Response · Authors · 2025-12-04
>
> # Comment to AC
>
> Dear AC,
>
> We would like to thank you for your hard work in overseeing the review process for our paper. During the rebuttal stage, we addressed several **misunderstandings** and **factual errors** in the reviewers’ comments:
>
> ---
>
> ### 1. Dynamically adjusting hash functions and privacy budgets may leak privacy
> *(Reviewer kZsJ, Q7gb)*
>
> This is a **misunderstanding**. We clarify that there is **no privacy leakage** in DLDP-BF in this scenario. Such adjustments **do not compromise privacy** because **all elements are locally perturbed under LDP** before being transmitted to the server. The server only observes the perturbed outputs and has no access to the internal state of the Bloom Filter.
> Consequently, even if **more hash functions** or a **larger privacy budget** are allocated to high-importance elements, the released outputs remain **strictly bounded by LDP**. An adversary cannot infer the importance of any individual element from the perturbed outputs—this guarantee follows directly from the definition of **differential privacy**.
>
> ---
>
> ### 2. Dynamic adjustment requires storing a large amount of additional information, violating the efficiency of Bloom Filters
> *(Reviewer kZsJ)*
>
> This is a **factual error**. Dynamic adjustment **does not undermine the efficiency** of Bloom Filters. Our DLDP-BF method, consistent with prior work (Jinyuan Jia. INFOCOM 2019. Terrance Liu. PMLR 2021. Fei Wei. VLDB 2024.), assumes access to **prior information or statistical distributions** when optimizing Bloom Filters, and therefore **does not require storing any sensitive element-wise data**.
>
> The additional information used for dynamic adjustment is **solely for parameter computation** and does not alter the **size, structure, or compactness** of the Bloom Filter. As a result, the **efficiency and lightweight nature** of the Bloom Filter remain fully preserved.
>
> ---
>
> ### 3. Theoretical and Experimental analysis
> *(Reviewer wb3W, Q7gb)*
>
> Following the reviewer’s suggestions, we added new theoretical and experimental results demonstrating the impact of varying **ε** and **m** on **RMSE** and **accuracy**. These results confirm **DLDP-BF's superior budget data utility** compared to baseline methods.
>
> Furthermore, we explicitly state that DLDP-BF is **theoretically equivalent** to existing methods in terms of **time and space complexity**.
>
> ---
>
> ## Summary
>
> We want to emphasize that **all reviewers acknowledged the theoretical contributions and practical validation** of our methodology, as follows:
>
> 1. The proposed **DLDP-BF** breaks through the limitation of traditional methods that use **uniform parameters for all elements** by dynamically adjusting the **number of hash functions** and the **privacy budget**. Notably, our DLDP-BF achieves **higher data utility**.
>
> 2. In theory, we prove that: (1) Our algorithm is comparable to existing Bloom Filters in terms of **time and space complexity**(Thm. 1 and 2); (2) our algorithm achieves **local differential privacy** (Thm. 3); (3) We prove that our algorithm achieves the **higher query accuracy utility** subject to privacy constraints the same as SOTA (Thm. 4).
>
> 3. Notably, the experimental results on multiple **real-world datasets** show that DLDP-BF **significantly outperforms existing methods** (Ke, CoRR, arXiv:2502.00693, 2025.; Ulfar, VLDB. 2014.). DLDP-BF achieves **higher query accuracy** and **lower RMSE**.
>
> ---
>
> Some of the **not-high scores** are based on **misunderstandings** and **incomplete experimental and theoretical analysis**. As we made **substantial improvements** to the revised manuscript based on these comments, we **appeal for a fair re-evaluation** that gives more consideration to our responses and the revised manuscript.

---

### Meta-Review · Area_Chair_WH4H · 2025-12-13

**Summary:**

Reviewers raised a number of concerns about this work. For example, the second concern from Reviewer kZsJ has some confusion about the storage cost v.s. the motivation of using Bloom filters, and Reviewer sE9T questions about the selection of the competitor RAPPOR. Overall, this paper makes some meaningful contributions on using prior statistical information to enhance the accuracy of LDP membership checker; however, it is unclear how significant the technical contribution is to the area of LDP (and a sub-class of frequency queries), and the paper can be motivated better to avoid the confusions from reviewers. So I would encourage the authors to revise it accordingly for future submission

**Reviewer Concerns:**

Most reviewers didn't reply after the rebuttal. Some concerns have been addressed, such as Q2 from Reviewer kZsJ, and W1 from Reviewer sE9T

**Reviewer Scores:**

Most reviewers didn't reply after the rebuttal.

---

### Decision · Program_Chairs · 2026-01-26

Reject